# MMP release following cartilage injury leads to collagen loss in intact tissue: A computational study

Moustafa Hamada[1]*, Atte S. A. Eskelinen[1], Joonas P. Kosonen[1], Cristina Florea[1], Alan J. Grodzinsky[2], Petri Tanska[1,3], Rami K. Korhonen[1]

1 Department of Technical Physics, University of Eastern Finland, Kuopio, Finland, 2 Departments of Biological Engineering, Electrical Engineering and Computer Science, and Mechanical Engineering, Massachusetts Institute of Technology, Cambridge, Massachusetts, United Stated of America, 3 Department of Oncology, Kuopio University Hospital, Wellbeing Services County of North Savo, Kuopio, Finland

* moustafa.hamada@uef.fi

## Abstract

Collagen damage in articular cartilage plays a key role in post-traumatic osteoarthritis, but the underlying mechanobiological pathways leading to collagen fibril degeneration after injury remain incompletely understood. We hypothesized that mechanical injurious loading induces localized cellular damage in cartilage, which in turn triggers the release of collagen-degrading matrix metalloproteinases (MMPs) and depth-wise collagen loss. To investigate this, we developed a computational mechano-signaling model for injured bovine cartilage, in which injury-induced cell damage is caused by excessive localized shear strains, leading to downstream MMP release, and spatially heterogeneous collagen degradation. The model predictions were compared to *ex vivo* cartilage explant experiments over 12 days post-injury. By day 12, the simulated bulk and depth-wise collagen loss aligned with our experimental findings quantified via Fourier-transform infrared microspectroscopy imaging (~30% average loss in the model vs. ~35% in the experiment). The results suggest that injury-induced cell damage and the downstream MMP activity can partly explain the depth-wise collagen content loss observed in the early days after cartilage injury. Ultimately, combining the current mechanistic approach with joint-level computational models could enhance the prediction of the onset and progression of cartilage degeneration following joint trauma.

## Author summary

Joint injuries that damage articular cartilage can increase the risk of developing post-traumatic osteoarthritis, a disease that progressively impairs joint function years after injury. Cartilage is a soft, load-bearing tissue that allows the smooth

**Data availability statement:** The generated data, codes, and models underlying this work are available via the Finnish Fairdata research services: https://etsin.fairdata.fi/dataset/3d36007a-be79-4a5b-b090-c380a59e3c4c.

**Funding:** This work was supported by the Research Council of Finland (grant number 363459 to RKK; partial salary to JPK and grant number 354916 to PT; partial salary to PT), the Novo Nordisk Foundation (grant number NNF21OC0065373 to RKK; partial salaries to MH, ASAE and RKK), the State Research Funding for university-level health research, Kuopio University Hospital, Wellbeing Service County of North Savo (grant number 5203080 to PT; partial salary to PT), the Sigrid Jusélius Foundation (to RKK; partial salary to MH), the Maire Lisko Foundation (to PT), the Päivikki and Sakari Sohlberg Foundation (grant number 240074 to PT), and the Finnish Ministry of Education and Culture's Pilot for Doctoral Programmes (Pilot project Mathematics of Sensing, Imaging and Modelling, partial salary to ASAE). The funders had no role in study design, data collection and analysis, decision to publish, or preparation of the manuscript.

**Competing interests:** The authors have declared that no competing interests exist.

articulation between bones, and its mechanical function depends strongly on collagen fibrils. After injury, collagen in cartilage can break down, but the links between mechanical injury, cell damage and collagen loss are not yet fully understood. In this study, we used a computational model to investigate how mechanical injurious loading contributes to enzymatic collagen degeneration in cartilage. The model shows that high mechanical shear strains can damage cartilage cells, which then release matrix metalloproteinases that gradually degrade collagen over time. Model results closely matched the experimental measurements of depth-dependent collagen content over a 12-day period following injurious loading. This work improves our understanding of how mechanically-driven cell damage can trigger enzymatic collagen loss in cartilage. It also supports the development of computational tools aiming to predict cartilage degeneration and osteoarthritis progression as well as evaluate potential treatments to limit protease-mediated tissue damage.

## 1. Introduction

Post-traumatic osteoarthritis (PTOA) is a subtype of osteoarthritis that can develop in response to joint injuries, such as anterior cruciate ligament injury, meniscus tear, or intra-articular fracture [1]. While timely clinical intervention after injury can restore joint function, altered biomechanics, prolonged inflammatory response, and compositional changes in articular cartilage tissue contribute to the long-term risk of developing PTOA [1–3]. PTOA can have devastating consequences, particularly for active and athletic individuals [3]. Symptoms often emerge years after the initial injury, presenting as joint instability, limited range of motion, and narrowing of the joint space [2,3]. In advanced stages, the disease may result in joint pain or complete disability, eventually necessitating joint replacement [2]. A key factor in advancing interventions for this disease is a detailed understanding of cartilage mechanobiology and the elucidation of mechanisms that govern cell–tissue interactions in the early stages following injury. Multiple experimental and computational works suggest that cartilage degeneration is initiated by cellular cues that lead to cartilage matrix loss [4–7]. However, the link between early-stage cell damage after injury and the degradation of cartilage collagen fibrils, as well as the spatial and temporal progression of this degradation within the tissue, remains ambiguous.

In response to injury, cell damage in cartilage can be represented by mitochondrial dysfunction, oxidative stress, and increased catabolic activity [7–9]. On the tissue level, injured cartilage can be characterized by aggrecan loss and collagen fibril network degradation [10–13]. This cascade of cellular and tissue-level responses can be interconnected through mechano-signaling pathways [14–16]. *Ex vivo* investigations have provided strong evidence that injury-induced cell damage, the imbalance between catabolic and anabolic chondrocyte activities, and increased production of matrix metalloproteinases (MMPs) and aggrecanases could later lead to enzymatic

degradation of collagen fibril network and aggrecan loss, respectively [7,9,17–19]. Specifically, elevated mechanical shear strains within the tissue can be co-localized with cellular damage such as mitochondrial dysfunction, production of reactive oxygen species, and cell apoptosis [4,9,20,21]. The high cellular strains can activate mechano-signaling pathways through cell surface receptors, such as integrins, and primary cilia [14,22,23]. This may culminate in the upregulation of MMPs which can be detected with immunohistochemical staining [24] (Fig 1B). The secreted MMPs, such as MMP-1, MMP-8, and MMP-13 [19,25], can cleave intact and mechanically ruptured collagen fibrils by binding to specific sites within the collagen triple helix structure, generating three-quarter and one-quarter length fragments [26]. These collagen fragments can then undergo further degradation by other MMPs such as MMP-2 and MMP-9 [27]. Ultimately, this cascade can lead to collagen content loss [13,19,23,27,28] which is observable along tissue depth and near chondral lesions as early as 12 days from injurious loading [24]. On the other hand, these catabolic protease activities can also be downregulated with several inhibitory or anabolic factors such as the increased expression of tissue inhibitors of metalloproteinases (TIMPs) [23] or blocking of MMPs from binding to collagen fibrils cleavage sites by aggrecan molecules [29].

Mechanobiological computational models offer a promising avenue for investigating the driving mechanisms behind early PTOA-related compositional changes, which are challenging to attain solely by experiments [28,30–32]. Regarding collagen damage, a major focus of current modeling studies is typically on the collagen mechanics such as collagen network softening/damage driven by excessive tensile stresses in the cartilage matrix or tensile strains in the individual collagen fibrils [33,34]. On the other hand, several computational studies have addressed enzymatic activity and biochemical signaling caused by traumatic joint injury that could lead to the loss of collagen fibrils [8,35–37]. However, these models have not incorporated cell damage-driven MMP-activity post-injury, which contributes to the localization and temporal progression of collagen loss in cartilage [28,30,36].

In this work, we developed a cell-driven mechano-signaling modeling framework to simulate depth-wise collagen content loss, motivated by our results from immunohistochemical staining of MMP-1 in injured cartilage explants (Fig 1B). We hypothesized that excessive shear strains trigger cell damage, leading to an increase in MMP production, and finally enzymatic cleavage of collagen fibrils [4,21]. To consider the protective effect of aggrecan against collagen cleavage (inhibited MMP-collagen binding), we also incorporated aggrecan loss due to aggrecanase-induced degeneration in our model [29,30]. To validate our approach, the simulations were compared against measured depth-wise collagen content in injuriously loaded young bovine cartilage explants on the day of the injury and 12 days after the injury [24].

## 2. Methods

### 2.1 Experiment design

This study utilizes a subset of data from previous experiments [24]. Briefly, cylindrical cartilage explants ($n = 63$, diameter = 3 mm, thickness = 1 mm) were collected from four different regions of the patellofemoral grooves harvested from two-week-old bovines ($N = 8$) [38,39]. A subset of explants was subjected to injurious loading (50% strain, 100%/s strain rate) to be assessed for collagen content on the day of injury (day 0, $n = 18$), and after 12 days of incubation in culture medium ($n = 6$). The remaining explants were kept as a free-swelling control group (day 0: $n = 18$ and day 12: $n = 21$).

### 2.2 Experimental collagen content assessment

Fourier-transform infrared microspectroscopy (FTIR, $5.5 \times 5.5$ μm$^2$ pixel size, 4 cm$^{-1}$ spectral resolution) was used to assess the spatial collagen content distribution [24]. Three 5-μm-thick unstained tissue sections were prepared per explant. The area under Amide I spectral region (wavenumber range: 1580 cm$^{-1}$ to 1720 cm$^{-1}$) was calculated pixel by pixel from each section per explant and used as an estimate for the collagen content (Fig 1A) [40,41]. Next, two full-depth and 200 μm wide regions of interest (ROI) were defined in each histological section. Collagen content was extracted from these ROIs, averaged perpendicular to depth, and normalized depth-dependently (0% = surface, 100% = bottom) to obtain

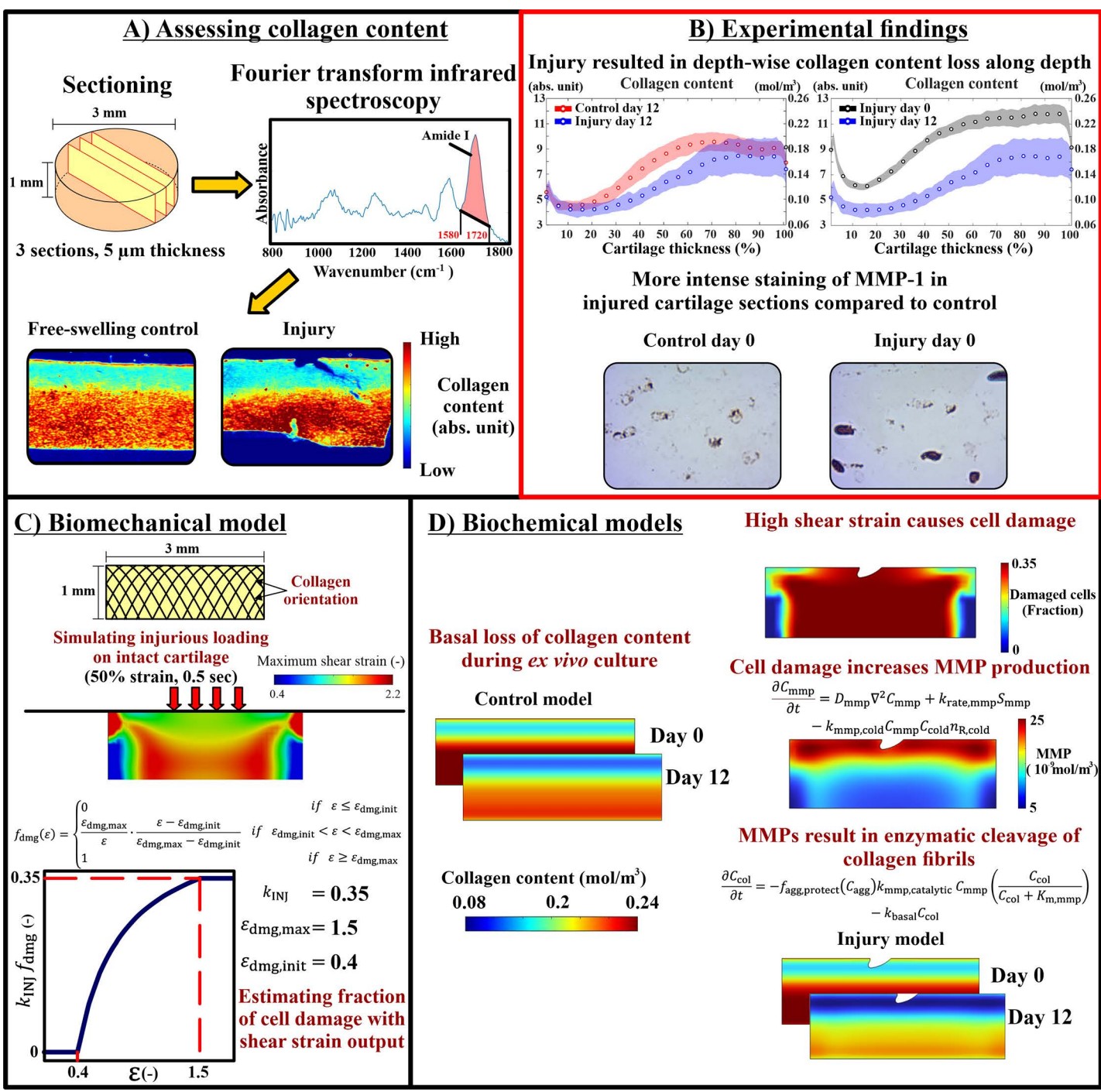

**Fig 1. Study workflow.** A) Collagen content was assessed experimentally using Fourier-transform infrared microspectroscopy in histological sections of control free-swelling and injuriously loaded cartilage explants. B) Key experimental findings showed that injurious loading can result in depth-wise loss of cartilage collagen content 12 days after injury when compared to free-swelling control group and injuriously loaded group on day 0. In addition, immuno-histochemical analysis revealed more intense staining of MMP-1 in injured cartilage compared to intact control. C) Injurious compression was simulated with finite element model to obtain maximum shear strain distribution. Then, the maximum shear strains were fed to damage function to estimate fraction of damaged cells. D) Collagen content loss was modeled over 12 days following injurious loading with reaction–diffusion equations. This biochemical model depicted cell damage due to high shear strains and subsequent catabolic response of increased production of matrix metalloproteinases (MMPs).

depth-wise collagen content profile for each explant. The ROIs were selected in intact areas located midway between the lesion and the section edges to ensure that the analyzed regions were unaffected by either the lesion or potential artifacts near the sample edges. A 200-µm width was chosen to capture the dominant depth-dependent gradient in collagen content while accounting for stochastic lateral variation in each section by averaging collagen content values across the 200-µm span.

Statistical analysis of the experimental data was performed using linear mixed-effects (LME) model implemented in IBM SPSS statistics 29.0 (IBM company, Armonk, NY, USA). LME model was used mainly to account for potential biological variation in the samples that are harvested from different anatomical regions and different animals. The experimental results are presented as mean depth-wise collagen content profiles with corresponding ±95% confidence intervals estimated from the LME models (Fig 1B) [24]. Additional details of the statistical analysis have been described in our previous work [24].

## 2.3 Biomechanical computational model

A 2D finite element (FE) model was created (Fig 1C) using ABAQUS software (v. 2023, Dassault Systèmes, Providence, RI, USA) to simulate injurious loading of an intact cartilage explant under unconfined compression. The 2D geometry of intact cartilage was created as in the experiment (diameter = 3 mm, thickness = 1 mm). Cartilage explant was meshed using 456 linear quadrilateral elements of type CPE4P. Prior to model implementation, different mesh densities were tested to ensure the model results were mesh-independent [42]. The cartilage was modeled as a fibril-reinforced porohyperelastic material with swelling, and the material parameters were obtained from the literature (Table 1) [39,43]. The orientation of the collagen fibrils was implemented with an arcade-like structure along depth based on previous bovine calf literature [39,44,45]. The initial depth-wise collagen fraction was implemented in the model based on the mean depth-wise collagen content in intact cartilage explants at day 0 (Table 1) [24]. Likewise, the depth-wise fixed charge density (FCD, assumed to correspond to the aggrecan content) distribution was implemented using optical density measurements from the same cartilage explants (characterized in [39]) and converted into FCD values as done previously [46]. The fluid fraction was obtained from literature [46]. All distributions were defined as depth-dependent, while lateral direction was assumed to remain homogeneous.

**Table 1. Main material parameters for the biomechanical model. z indicates normalized distance from the cartilage surface.**

| Material parameter | Value | Description | Reference |
|---|---|---|---|
| $E_f$ (MPa) | 20.0 | Fibril network modulus | [39] |
| $E_m$ (MPa) | 0.16 | Non-fibrillar matrix modulus | [43] |
| $v_{nf}$ (-) | 0.40 | Poisson's ratio of the non-fibrillar matrix | [47] |
| k ($10^{-15}$ m$^4$ N$^{-1}$ s$^{-1}$) | 1.3 | Hydraulic permeability | [43] |
| Distributions | Value | Description | Reference |
| $n_{f,0}$ (-) | $0.85 - 0.1z$ | Initial depth-wise fluid fraction | [39] |
| $c_{col,0}$ (-) | $-187.1z^6 + 328.6z^5 + 22.9z^4 - 363.4z^3 + 247.8z^2 - 47.9z + 8.782$ | Initial depth-wise collagen fraction | [24] |
| $c_{FCD,0}$ (mEq· ml$^{-1}$) | $-4.4z^6 + 15.2z^5 - 21.0z^4 + 14.9z^3 - 5.8z^2 + 1.1z + 0.03$ | Initial depth-wise fixed charge density | [39] |

**2.3.1 Boundary and loading conditions.** The bottom surface of the explant was fixed in the axial direction, while a single node at the intersection point between the center line of the explant and the bottom surface was fixed in the lateral direction to fix the model in place but allow lateral expansion of the tissue simultaneously. The contact between the tissue's top surface and the compressing plate was modeled as a frictionless surface-to-surface interaction. Free fluid flow was allowed from the lateral edges (pore pressure was set to 0). Finally, similarly to the experiments, a single load-unload cycle with a 50% strain magnitude (strain rate 100%/s, 50% strain reached at 0.5 s) was applied on the top surface of the explant to simulate the injurious loading.

## 2.4 Biochemical computational model

A 2D geometry of cartilage explant was created with same dimensions as in the biomechanical model. We did not model lesion formation or crack propagation in the mechanical model, but lesion geometry was added to the biochemical model geometry by segmenting the lesion geometry from one histological section (Fig 1D). This lesion geometry was created to visually assess the localization of collagen loss around this region after injury.

**2.4.1 Governing equations.** To simulate cell damage and subsequent collagen loss over 12 days, the maximum shear strain distribution at 50% of compressive strain was obtained from the biomechanical model by calculating the maximum difference between the principal components ($\varepsilon_{p,1}$, $\varepsilon_{p,2}$, $\varepsilon_{p,3}$) of the Green–Lagrangian strain tensor [31]:

$$\varepsilon = \max\left\{\left|\varepsilon_{p,1} - \varepsilon_{p,2}\right|, \left|\varepsilon_{p,1} - \varepsilon_{p,3}\right|, \left|\varepsilon_{p,2} - \varepsilon_{p,3}\right|\right\}, \tag{1}$$

The maximum shear strain values were used as they have been linked with chondrocyte damage in deformed tissue regions [39]. Chondrocyte damage was modelled with a stepwise non-linear function $f_{dmg}(\varepsilon)$ [21] (Fig 1C):

$$f_{dmg}(\varepsilon) = \begin{cases} 0 & \text{if } \varepsilon \leq \varepsilon_{dmg,init} \\ \frac{\varepsilon_{dmg,max}}{\varepsilon} \cdot \frac{\varepsilon - \varepsilon_{dmg,init}}{\varepsilon_{dmg,max} - \varepsilon_{dmg,init}} & \text{if } \varepsilon_{dmg,init} < \varepsilon < \varepsilon_{dmg,max} \\ 1 & \text{if } \varepsilon \geq \varepsilon_{dmg,max}, \end{cases} \tag{2}$$

in which we assumed that cells start to turn damaged when the maximum shear strain value exceeds 40% ($\varepsilon_{dmg,init} = 0.4$) and after that, the amount of damaged cells increases non-linearly as a function of strain until the maximum fraction of damaged cells is reached at the maximum shear strain value of 150% ($\varepsilon_{dmg,max} = 1.5$). These threshold values are based on previous computational studies using fibril-reinforced material models and experimental data using excessive levels of strain (indentation test) [9,39]. Subsequently, the concentration of damaged cells $C_{cell,damaged}$ was defined in COMSOL Multiphysics (version 6.1, MA, USA) as a fraction of the concentration of healthy cells $C_{cell,healthy}$:

$$C_{cell,damaged} = k_{INJ} \, f_{dmg}(\varepsilon) \, C_{cell,healthy}, \tag{3}$$

where $k_{INJ} = 0.35$ is a multiplier to limit the maximum fraction of cell damage in the model to 35% from healthy cells. This estimate of cell damage is adopted from previous cell apoptosis measurements in injured young bovine cartilage [25].

The temporal and spatial changes of the concentrations of species in cartilage post-injury were modeled with reaction–diffusion equations as [30]:

$$\frac{\partial C_i}{\partial t} = D_i \nabla^2 C_i \pm R_i, \tag{4}$$

where $C_i$ indicates the concentration of a species $i$. Species considered in the model are intact collagen (col), degraded collagen (cold), intact aggrecan (agg), degraded aggrecan (aggd), MMP (mmp), and aggrecanase (aga). $D_i$ is the effective

diffusivity of the species $i$ while $R_i$ denotes a variable that encompasses the rates of production or degradation of the species $i$ [30]. We only considered the net effect of proteases and different types of MMPs (such as MMP-1, and MMP-3) and aggrecanases (such as ADAMTS-4 and ADAMTS-5) for model simplicity. The model accounts for aggrecan loss, along with the inclusion of degraded collagen and degraded aggrecan, as these components influence MMP binding to intact collagen fibrils and contribute to the regulation of collagen loss [28,29].

As the activation and secretion of MMPs and aggrecanases after injury is not instantaneous, time-dependent stimulus variables $S_{mmp}$ and $S_{aga}$ were introduced to represent the time delay of MMP and aggrecanase activations [30]:

$$\frac{\partial S_j}{\partial t} = \alpha_j \left( k_j C_{cell,damaged} - S_j \right),$$

(5)

where $\alpha_j$ corresponds to the rate constant for the net stimulus of protease $j$ ($j$ = mmp or aga), and $k_j$ is a constant for protease released from damaged cells. The subsequent change in MMP production over time was governed by:

$$\frac{\partial C_{mmp}}{\partial t} = D_{mmp}\nabla^2 C_{mmp} + k_{rate,mmp}S_{mmp} - k_{mmp,cold}C_{mmp}C_{cold}n_{R,cold},$$

(6)

where $k_{rate,mmp}$ is the rate constant for generating MMPs based on damaged cell-driven stimulus, $k_{mmp,cold}$ is a rate constant for MMP binding to degraded collagen and $n_{R,cold}$ represents the number of MMP binding sites on degraded collagen [28]. The model assumes that the release of MMPs occurs at the same rate over the 12-day simulation. To simulate changes in collagen content, the model considered two main factors (Fig 1D): MMP secretion due to cell damage and the protection of collagen by aggrecan [30]. The protection by aggrecan is based on previous suggestions that the presence of aggrecan can hinder the binding of MMPs to their binding sites on collagen fibrils [29,30]. The change in collagen concentration was described as:

$$\frac{\partial C_{col}}{\partial t} = -f_{agg,protect}(C_{agg})k_{mmp,catalytic}\,C_{mmp}\left(\frac{C_{col}}{C_{col} + K_{m,mmp}}\right) - k_{basal}C_{col},$$

(7)

where $k_{mmp,catalytic}$ defines the rate of MMP catalytic activity, $K_{m,mmp}$ represents Michaelis constant for MMPs which describes the MMP concentration at which the MMP–collagen-binding reaction occurs at half-maximum rate [30], and $k_{basal}$ describes the constant rate of basal loss of collagen content (spontaneous decrease in collagen content as observed in free-swelling explants between days 0 and 12). This basal loss accounts for collagen degradation that may result from explant culturing rather than injury-driven enzymatic degradation [48]. $f_{agg,protect}$ is a function that defines how local aggrecan concentration limits the tendency of MMPs to cleave collagen fibrils [30]:

$$f_{agg,protect}(C_{agg}) = \frac{1}{1 + \left(\frac{C_{agg} + C_{aggd}}{k_{act}}\right)^{-n}},$$

(8)

where $C_{agg}$ and $C_{aggd}$ are the concentrations of intact and degraded aggrecan respectively, $k_{act}$ defines aggrecan concentration at half-maximal MMP activity, and $n$ is the Hill coefficient of MMP activity [30].

In the model representing the free-swelling control cartilage explants, collagen loss was assumed to occur only due to the constant rate of basal loss $k_{basal}$ (see Fig 1D). Please see S1 Text for more details on the equations governing aggrecanase and aggrecan concentrations (Eq. S1-S3 in S1 Text), as well as the parameter values used in the model (Table A in S1 Text).

**2.4.2 Boundary and initial conditions.** For all the variables, zero flux boundary conditions were defined on the bottom surface of the explant, assuming no mass transfer to occur across this surface. For intact aggrecan, degraded

aggrecan, and MMPs, Robin boundary conditions were defined through the top and lateral surfaces (representing concentration gradient normal to the surface interface) [30]. For degraded collagen and aggrecanases, Dirichlet boundary conditions were defined at these same surfaces (zero concentration) [30]. The initial healthy cell concentration before the simulation was assumed spatially homogeneous, as in immature bovine articular cartilage [30]. The damaged cell concentration was described in Equation (3), and the concentrations for MMPs, aggrecanases, degraded collagen, and degraded aggrecan were assumed to be zero as initial conditions.

The initial depth-wise collagen concentration was implemented based on scaling the collagen content absorption unit obtained from the FTIR measurements ($C_{col,0}$, Table 1) to match with the mean collagen concentration of 0.2 moles/m$^3$ reported in Kar et al. [30]:

$$C_{col,init} = \frac{C_{col,0}}{46.2276}[\text{mol/m}^3]$$

(9)

The initial aggrecan distribution was obtained from the fixed charge density (Table 1) as [39]:

$$C_{agg,init} = \frac{c_{FCD,0} \cdot 502.5}{-2 \times 2.5 \cdot 1000}[\text{mol/m}^3]$$

(10)

## 2.5 Sensitivity analysis

A sensitivity analysis was conducted for parameters lacking quantitative experimental data but significantly influencing collagen loss in the model. These included parameters governing the extent of cell damage ($k_{INJ}$ and $\varepsilon_{dmg,max}$), and parameters regulating the enzymatic cleavage of collagen by MMPs ($k_{rate,mmp}$, $k_{act}$, $k_{mmp}$, and $k_{mmp,catalytic}$). Reference values for these parameters were selected based on preliminary simulations (Table 2), in which the simulated depth-wise collagen content was matched with the experimentally measured data [24]. For all parameters, a uniform ±50% range of values was applied to systematically assess the sensitivity of collagen content to changes in each parameter. This ±50% range represents a conventional modeling assumption based on preliminary calibration, given the absence of direct experimental measurements for these parameters. During preliminary calibration, we observed that $k_{act}$ has a limited influence on collagen loss under the current aggrecan concentration used in the model. Therefore, we further evaluated its effect by increasing the value by a factor of 100.

**Table 2. Biochemical model parameters used for sensitivity analysis. Bolded values represent the reference value. Most of the parameters were analysed with ±50% value range. $k_{act}$ was also tested with 100 times higher value to characterize its effect since no difference was observed within the ±50% range.**

| Material Parameter | Range | Description | Reference |
|---|---|---|---|
| $k_{INJ}$ (−) | 0.2, **0.35**, 0.5 | Maximum fraction of healthy cells turning damaged after injury (Eq 3) | [7] |
| $\varepsilon_{dmg,max}$ (%) | 100%, **150%**, 200% | Maximum shear strain threshold for maximum cell damage (See Fig 1c) | [21], model fit |
| $k_{rate,mmp}$ ($10^{-5}$ s$^{-1}$) | 1.8, **3.6**, 5.3 | Rate constant for MMPs generation from damaged cells (Eq 6) | model fit, [30] |
| $k_{act}$ ($10^{-5}$ mol m$^{-3}$) | 1.5, **3**, 4.5, 300 | Aggrecan concentration at half-maximum of MMP activity (Eq 8) | [30] |
| $k_{mmp}$ ($10^{-21}$ mol) | 0.17, **0.35**, 0.52 | Production of MMPs by damaged cells (Eq 5) | model fit |
| $k_{mmp,catalytic}$ (s$^{-1}$) | 0.75, **1.5**, 2.25 | Catalytic activity rate of MMPs with collagen fibrils (Eq 7) | [30] |

In addition, to address the experimentally observed lateral heterogeneity in collagen content distribution, separate sensitivity analyses were conducted in which lateral variation in the initial collagen content distribution was introduced into the model. The methods and results of these analyses are described in detail in Section D in <u>S1 Text</u>.

## 3. Results

The simulated control model predicted on average 19% decrease in the bulk tissue collagen content over 12 days (<u>Fig 2A</u> & <u>2C</u>). In the control group explants, the bulk tissue collagen content was on average 20% (±6%) smaller on day 12 compared to day 0 (<u>Fig 2B</u> & <u>2C</u>).

The simulated injury model predicted an average 30% (±7%) bulk decrease in collagen content (<u>Fig 2D</u> & <u>2F</u>). On the other hand, the experimental injury group on day 12 showed an average bulk decrease of 35% (±6%) compared to day 0 (<u>Fig 2E</u> & <u>2F</u>). In these simulations for injury, depth-wise estimates for collagen content were mostly within the 95% confidence interval of the experimental results (at 0–30% and 60–100% of normalized tissue depth, <u>Fig 2F</u>).

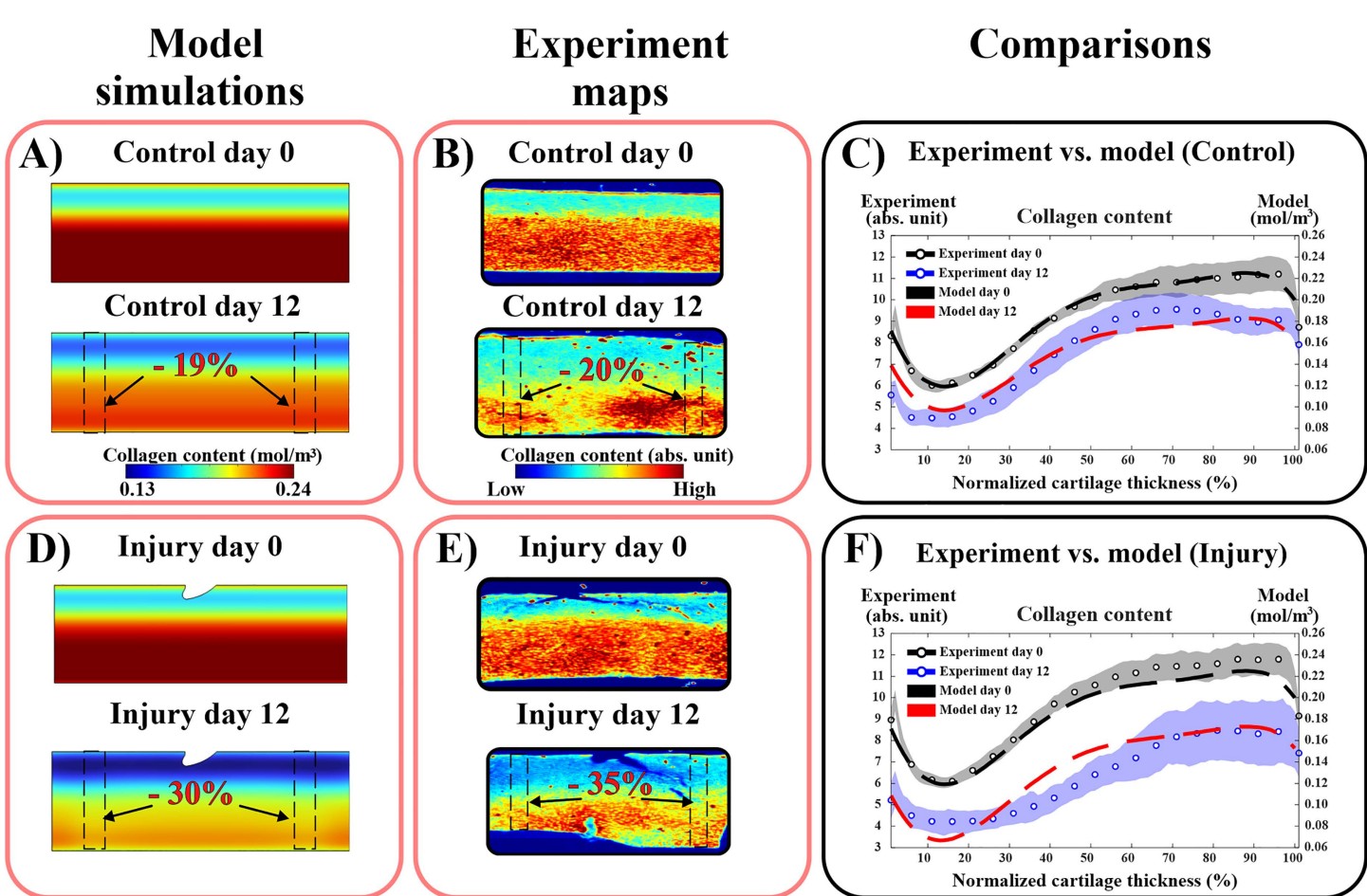

**Fig 2. Comparison of experimental and simulated depth-wise collagen content loss.** A)&B) collagen distribution overtime in the free-swelling control cartilage in the model and experiment, respectively. C) comparison between model and experimental results along 1 mm-thickness control cartilage showing average 19% bulk decrease in collagen content over 12 days in the model compared to 20% decrease in the experiment. D)&E) collagen distribution overtime in the injuriously loaded cartilage in the model and experiment, respectively. F) the injury model matched with the experimental depth-wise collagen content on day 12, showing on average 30% bulk decrease in collagen content by day 12 compared to 35% decrease in the experiment. The experimental findings are presented as mean ±95% confidence intervals.

Sensitivity analysis done on the injury model revealed that increasing or decreasing the shear strain threshold for maximum cell damage (from $\varepsilon_{dmg,max} = 150\%$ to $\varepsilon_{dmg,max} = 200\%$ or $100\%$) had a negligible effect on collagen loss (Fig 3A). On the other hand, increasing the maximum allowed cell damage (from $k_{INJ} = 0.35$ to $k_{INJ} = 0.5$) decreased collagen content on day 12 by ~15% (from 30% to 45% decrease) mostly in the superficial region (at 0–22% of the normalized tissue depth, Fig 3B).

Increasing the rate of MMP generation by damaged cells from $k_{rate,mmp} = 3.6 \times 10^{-5}$ 1/s to $k_{rate,mmp} = 5.3 \times 10^{-5}$ 1/s and MMP catalytic activity rate from $k_{mmp,catalytic} = 1.5$ 1/s to $k_{mmp,catalytic} = 2.25$ 1/s decreased collagen content in both cases on average by 17% (from 30% to 47% decrease) with the most loss occurring at the normalized tissue depth of 0–20% (Fig 4A & 4B). Increasing the amount of MMPs released from damaged cells from $k_{mmp} = 0.35 \times 10^{-21}$ mol to $k_{mmp} = 0.52 \times 10^{-21}$ mol decreased collagen content by 18% (from 30% to 48% decrease) at 0–25% of the normalized tissue depth (Fig 4C). On the other hand, increasing the parameter for aggrecan concentration at half-maximum of MMP activity from $k_{act} = 3 \times 10^{-4}$ mol/m³ to $k_{act} = 4.5 \times 10^{-4}$ mol/m³ (Equation 8) did not affect the collagen concentration over 12 days (Fig 4D). However, a considerable increase in collagen content by ~8% at 0–10% of the normalized tissue depth starts when the parameter value was increased from $k_{act} = 3 \times 10^{-5}$ mol/m³ to $k_{act} = 3 \times 10^{-3}$ mol/m³.

Results reflecting the effect of lateral variation in collagen content distribution are presented in Figs D and E in S1 Text.

## 4. Discussion

We created a computational model to simulate cell damage and MMP release caused by injurious loading, and the resulting loss of collagen content in bovine articular cartilage explants over 12 days post-injury. We hypothesized that cell damage in injured cartilage is pronounced in highly strained tissue regions, leading to the activation of MMPs, which subsequently degrade collagen fibrils and result in collagen content loss. The injury computational model predicted depth-wise collagen content loss, consistent with experimentally observed data over 12 days (Fig 2) [24]. Our simulations

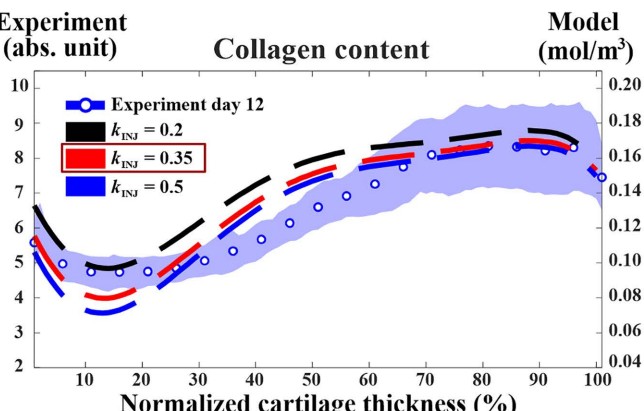

**Fig 3. Sensitivity analysis for parameters controlling cell damage in the injury model.** A) Increasing or decreasing the shear strain threshold for maximum cell damage by 50% had negligible effect on depth-wise collagen content. B) Increasing the maximum fraction of healthy cells turning damaged ($\mathbf{k_{INJ}}$) from reference value of 0.35 to 0.5 resulted in increased collagen content loss by ~15% in the superficial region of cartilage (~0–22% of normalized depth) while decreasing the value to 0.2 decreased the loss by ~25% in the superficial cartilage. The red color highlights the parameter value used in the reference model. The experimental findings are presented as mean ±95% confidence intervals.

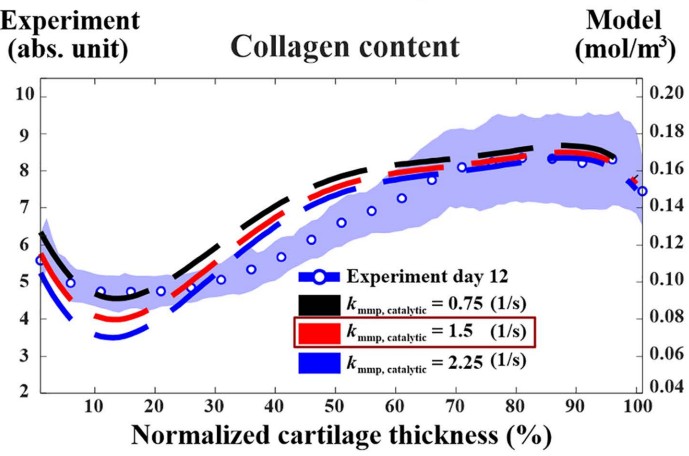

## A) Catalytic activity rate of MMPs with collagen fibrils

## B) Rate constant for MMPs generation from damaged cells

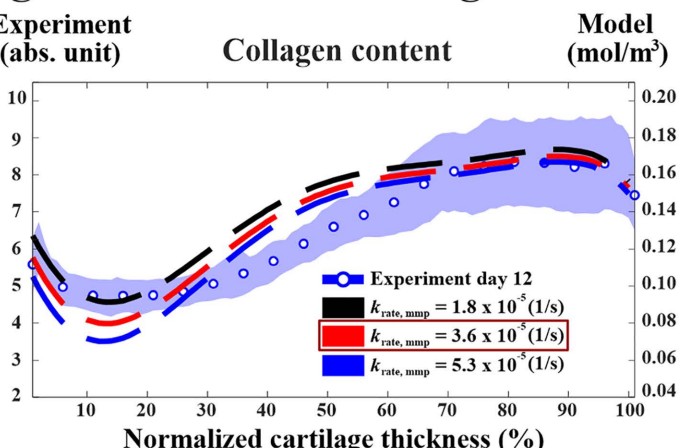

## C) Production of MMPs by damaged cells

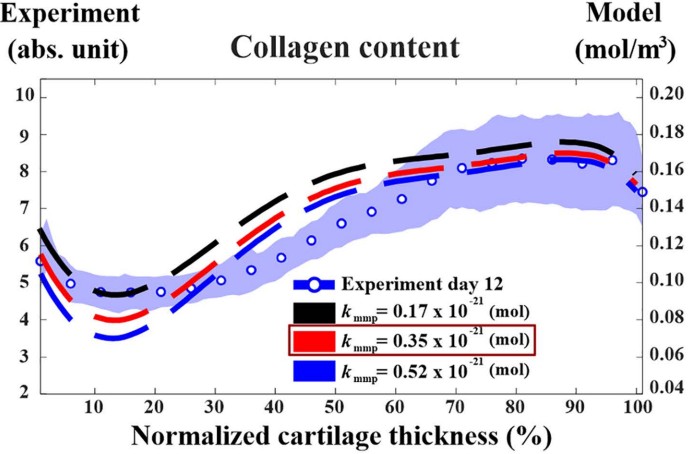

## D) Aggrecan concentration at half-maximum of MMP activity

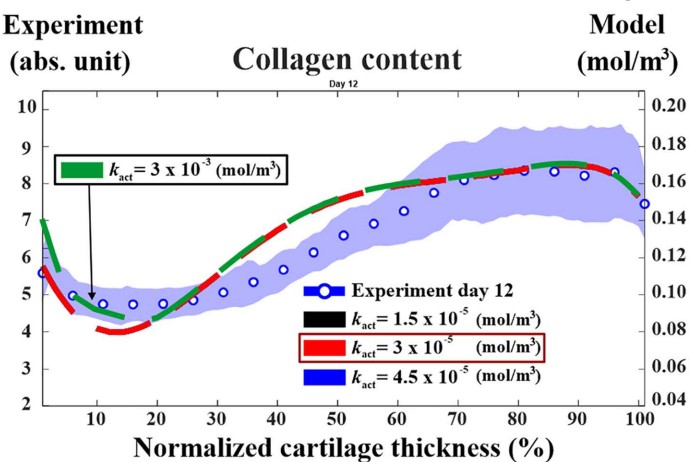

**Fig 4. Sensitivity analysis for parameters controlling production and enzymatic activity of MMPs.** A)&B) increasing the MMP catalytic rate ($k_{mmp,catalytic}$) by 50% and rate constant for MMPs generation from damaged cells ($k_{rate,mmp}$) parameters increased the loss of collagen content similarly by ~17% with the most loss in the superficial region (~0–20% of normalized depth) as compared to reference model (highlighted in red color). C) model with increased production of MMPs by damaged cells $k_{mmp}$ exhibited ~18% more collagen content loss (~0 to 25% of normalized depth) compared to reference model. D) changing aggrecan concentration at half-maximum of MMP activity ($k_{act}$) parameter had negligible effect on the depth-wise collagen content. However, increasing the value by 100 times more instead of 50% resulted in higher collagen content by ~8% on average compared to reference model at 0–10% of normalized tissue depth. The experimental findings are presented as mean ±95% confidence intervals. MMP = matrix metalloproteinase.

suggest that catabolic cellular activity and the upregulation of MMP activity play a key role in collagen loss during the early stages of cartilage injury.

The experimental results of the free-swelling control group showed a significant decrease in collagen content on day 12 throughout the entire depth of the tissue compared to day 0 (Fig 2C). Similar observations were reported in previous *ex vivo* studies, where aggrecan loss and cell death were observed in free-swelling cartilage within 12 days [6,38,49,50].

Therefore, this spontaneous basal collagen loss was considered in our workflow with the rate constant $k_{basal}$. Including spontaneous collagen loss in the injury model facilitated the differentiation between collagen loss caused in the free-swelling culture conditions and collagen loss caused by the injury-induced MMP release (Fig 2F). Detailed comparison of the injury model with and without the basal loss term are available in Section B in S1 Text for further reference.

The presented injury model successfully matched with the experimentally observed decrease in depth-wise collagen content across the superficial and deeper tissue regions over 12 days. Thus, our proposed mechanism of injury-related MMP release and enzymatic cleavage of collagen can explain the collagen loss observed experimentally in injured cartilage [24]. This finding aligns with previous experimental and computational studies investigating collagen fibril damage early after excessive and injurious loading [36,51]. Recent computational frameworks have also explored collagen and extracellular matrix loss in cartilage after overloading with attention to the cellular catabolic response, biochemical signaling, and increased proteolytic activity [8,28,36]. For example, Rahman et al. [36], using 3D FE model, captured significant bulk collagen loss (~10%) over 3 months due to overloading and MMP-driven degradation. Our model exhibits a more rapid activation of MMPs and collagen loss, as it is based on *ex vivo* experimental data from bovine cartilage subjected to controlled injurious loading protocol. Rahman et al.'s framework, however, relied on *in vivo* data from human subjects, where cartilage degeneration typically occurs over a much longer timeframe compared to the controlled *ex vivo* environment [36]. On the other hand, Kapitanov et al.'s mathematical model addressed the role of blunt impact loading (drop tower impact loading of 2.18 J/cm$^2$ energy) on cell damage and the release of interleukin-6 (IL-6) inflammatory cytokines [35]. Their model results showed an increase in IL-6 and reactive oxygen species close to the loaded regions, though it did not capture the extracellular matrix loss over a 14-day period. Notably, their model used axial strain to estimate cell damage, which could yield different quantifications of damaged cells. Furthermore, they used homogeneous linear elastic mechanical properties for cartilage with isotropic matrix distribution, which may lead to different mechanical and chemical tissue responses compared to our models that incorporated depth-dependent distribution of tissue structure.

The predicted collagen content in our model deviated from the 95% confidence interval of the experiments at the middle region of the explant (~30–60% of tissue thickness, Fig 2B). This suggests that additional mechanisms may also contribute to collagen loss induced by injurious loading. For instance, previous experimental studies have proposed that collagen fibril damage could be mechanically induced by excessive loading, resulting in fibrils rupture and subsequent loss of collagen content [52,53]. Future work with our models could incorporate this hypothesis and test the validity of this mechanism by including collagen fibrils strain into the simulations. Also, this deviation may stem from some of our model's limitations. First, our model did not discriminate between the various types of MMPs, each with distinct diffusion coefficients and enzyme kinetics, nor did it explicitly consider spatiotemporal variations in the concentrations of tissue-inhibitors of metalloproteinases. Additionally, we did not discern between different types of cell damage, such as cell apoptosis, mitochondrial dysfunction, or oxidative stress, which could locally contribute to collagen degradation [17,54,55]. Furthermore, the biomechanical model relies on literature-based mechanical properties of young bovine cartilage rather than properties derived from our experimental samples which may vary between the samples [56,57]. These variations in mechanical properties could contribute to the observed deviation between the model and experimental results.

Our sensitivity analysis revealed that the parameters responsible for the fraction of cell damage ($k_{INJ}$) and the production and activity of MMPs ($k_{rate,mmp}$, $k_{mmp,catalytic}$, and $k_{mmp}$) substantially influence the numerical collagen content estimates [13,23,40,58]. However, ± 50% variation in $k_{act}$ (i.e., aggrecan concentration at half-maximum of MMP activity) yielded no differences in the depth-wise collagen content. We adopted the same parameter value as Kar et al. [30], who demonstrated in their model that a sufficiently high concentration of aggrecan can shield the collagen network from enzymatic degradation caused by IL-1-driven MMP release. However, our study utilized a lower aggrecan concentration (~half the concentration) compared to their model, potentially limiting the collagen-protecting effect of aggrecan. On the other hand, increasing of $k_{act}$ parameter 100 times the reference value resulted in substantially higher collagen content compared to the reference value (Fig 4D) suggesting that further calibration of this parameter is needed. This could enhance our

understanding of the mechanistic relationship between aggrecan and collagen in either aggravating or protecting cartilage tissue from the degradation [29]. Also, this could open new avenues for investigating whether potential interventions that target aggrecan biosynthesis in cartilage would play a role in limiting the enzymatic degradation of collagen fibrils [29,38,55].

Our model was developed using immature bovine cartilage, which has different material, mechanical, and compositional properties compared to mature human cartilage. These differences, along with the specific loading protocol applied in this study (in terms of strain amplitude and strain rate), may influence the extent of cell damage and the subsequent mechano-inflammatory response. Therefore, the experimental findings and model parameter values presented here might not be directly translatable to human cartilage without further validation. In addition, the experimental collagen content data used in this work for comparison were available only at two time points (i.e., day 0 and day 12). Incorporating intermediate time points in future work would enable a more accurate characterization of the temporal progression of MMP upregulation and catabolic activity following injury-induced cell damage and thereby allow a more precise temporal assessment of collagen loss over the 12-day period.

We acknowledge that, in the context of traumatic joint injuries, collagen fibrils degeneration in cartilage may also be influenced by inflammatory responses originating from surrounding joint tissues, such as the synovium [26,59,60]. Injury to these tissues can trigger elevated expression of cytokines such as IL-1 and IL-6 [26,61], which may subsequently infiltrate the cartilage and stimulate MMP production [30]. Moreover, these cytokines can exacerbate degradation of collagen fibrils through additional MMP-independent pathways, such as suppressing collagen and aggrecan biosynthesis, increasing oxidative stress and promoting cell apoptosis. While our current model focuses on MMP secretion within cartilage in response to direct mechanical injurious loading, this framework can be readily extended to incorporate these exogenous inflammatory pathways. Integrating such mechanisms in future work would enable a more comprehensive simulation of the post-injury cartilage environment.

It is worth noting that the model accounted for the depth-dependent distribution of collagen content in cartilage, while a homogeneous distribution was assumed along the lateral direction. Experimentally, however, the samples exhibited stochastic and sample-specific lateral variation in collagen content distribution (Fig C in S1 Text). Accurately representing this variability would require sample-specific modeling, which was beyond the scope of this work. Nevertheless, when we introduced lateral variation in the initial collagen content distribution of the model, utilizing the sample with the largest x-axis heterogeneity (Figs D and E in S1 Text), this variation had only a minor effect on collagen loss.

Lastly, our model did not replicate the previously reported amplified loss of collagen content localized near lesions [24]. This localized collagen degradation was observed to occur rapidly after injury, coinciding with localized cell death on day 0 [24,38]. This discrepancy suggests the involvement of additional mechanisms driving the localized loss of collagen content, potentially including initial mechanically induced matrix loss [52,53].

In conclusion, our model replicates experimental collagen loss after injury through injury-induced mechano-signaling in cartilage cells, leading to downstream MMP activity and subsequent early depth-wise collagen loss. However, contributions from other biochemical pathways and biomechanically-driven factors may also play a role. With additional experimental data and further calibration, our model can be valuable for testing the effectiveness of treatment interventions aimed at limiting proteolytic collagen degeneration. This modeling framework could be also implemented into joint-level computational models designed to estimate the progression of PTOA and treatment efficacy [62].

## Supporting information

**S1 Text. Supplementary material file containing more detailed information regarding the models and further sensitivity analysis. Fig A.** Comparison of the injury model with and without the basal loss of collagen content. **Fig B.** Comparison of the biomechanical model with and without lesion geometry. **Fig C**. Chemical maps of collagen content. **Fig**

**D**. Effect of imposing lateral gradient in the initial collagen content distribution. **Fig E.** Effect of imposing lateral gradient in the initial cell damage. Table A. Biochemical model parameters.
(DOCX)

## Author contributions

**Conceptualization:** Moustafa Hamada, Atte S. A. Eskelinen, Joonas P. Kosonen, Cristina Florea, Alan J. Grodzinsky, Petri Tanska, Rami K. Korhonen.

**Data curation:** Moustafa Hamada.

**Formal analysis:** Moustafa Hamada.

**Funding acquisition:** Atte S. A. Eskelinen, Petri Tanska, Rami K. Korhonen.

**Investigation:** Moustafa Hamada.

**Methodology:** Moustafa Hamada, Atte S. A. Eskelinen, Joonas P. Kosonen, Cristina Florea, Alan J. Grodzinsky, Petri Tanska, Rami K. Korhonen.

**Project administration:** Petri Tanska, Rami K. Korhonen.

**Resources:** Alan J. Grodzinsky, Petri Tanska,Rami K. Korhonen.

**Supervision:** Atte S. A. Eskelinen, Petri Tanska, Rami K. Korhonen.

**Visualization:** Moustafa Hamada, Atte S. A. Eskelinen, Joonas P. Kosonen.

**Writing – original draft:** Moustafa Hamada.

**Writing – review & editing:** Moustafa Hamada, Atte S. A. Eskelinen, Joonas P. Kosonen, Cristina Florea, Alan J. Grodzinsky, Petri Tanska, Rami K. Korhonen.

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
