## [Decision Letter · Decision Letter 0]

25 Sep 2025

PCOMPBIOL-D-25-01150

MMP release following cartilage injury leads to collagen loss in intact tissue – a computational study

PLOS Computational Biology

Dear Dr. Hamada,

Thank you for submitting your manuscript to PLOS Computational Biology. After careful consideration, we feel that it has merit but does not fully meet PLOS Computational Biology's publication criteria as it currently stands. Therefore, we invite you to submit a revised version of the manuscript that addresses the points raised during the review process.

Please submit your revised manuscript within 60 days Nov 25 2025 11:59PM. If you will need more time than this to complete your revisions, please reply to this message or contact the journal office at ploscompbiol@plos.org. Please include the following items when submitting your revised manuscript:

We look forward to receiving your revised manuscript.

Kind regards,

Arne Elofsson

Section Editor

PLOS Computational Biology

**Journal Requirements:**

At this stage, the following Authors/Authors require contributions: Moustafa Anwar Hamada, Atte A Eskelinen, Joonas P Kosonen, Cristina Florea, Alan Grodzinsky, Petri Tanska, and Rami K Korhonen. Please ensure that the full contributions of each author are acknowledged in the "Add/Edit/Remove Authors" section of our submission form.

3) Some material included in your submission may be copyrighted. According to PLOSu2019s copyright policy, authors who use figures or other material (e.g., graphics, clipart, maps) from another author or copyright holder must demonstrate or obtain permission to publish this material under the Creative Commons Attribution 4.0 International (CC BY 4.0) License used by PLOS journals. Please closely review the details of PLOSu2019s copyright requirements here: PLOS Licenses and Copyright. If you need to request permissions from a copyright holder, you may use PLOS's Copyright Content Permission form.

Potential Copyright Issues:

i) Figure 1a. Please confirm whether you drew the images / clip-art within the figure panels by hand. If you did not draw the images, please provide (a) a link to the source of the images or icons and their license / terms of use; or (b) written permission from the copyright holder to publish the images or icons under our CC BY 4.0 license. Alternatively, you may replace the images with open source alternatives. See these open source resources you may use to replace images / clip-art:

ii) The following Figure contains a logo or branding: 1d. We are not permitted to publish this under our CC-BY 4.0 license, even with permission. We ask that you please remove or replace it.

4) Thank you for stating  "Upon acceptance of the manuscript for publication, the full set of metadata, codes and models will be made publicly available and the accession numbers/DOI will be provided (Fairdata Research Data Storage Service, https://etsin.fairdata.fi/)." We strongly recommend all authors deposit their data before acceptance, as the process can be lengthy and hold up publication timelines. Please note that, though access restrictions are acceptable now, your entire minimal dataset will need to be made freely accessible if your manuscript is accepted for publication. This policy applies to all data except where public deposition would breach compliance with the protocol approved by your research ethics board. 

**Reviewers' comments:**

Reviewer's Responses to Questions

Reviewer #1: Hamada’s work presents a finite element–based mechano-signaling model that mechanistically links acute mechanical injury to biochemical collagen degradation in articular cartilage. The model effectively captures key causal sequences: mechanical insult → cell damage → MMP upregulation → collagen degradation, which provides mechanistic insight that bridges the gap between mechanical trauma and subsequent matrix breakdown. Notably, authors validate their simulations against ex vivo data from injured bovine cartilage explants cultured for 12 days. They demonstrated strong concordance with model predictions (~30% simulated vs. ~35% experimental collagen loss), thereby reinforcing the biological plausibility of the proposed mechanism.

This study delivers a compelling and well-validated mechano-signaling model that advances the field of computational cartilage degeneration. Its integration of mechanical strain, cell viability, and enzyme-mediated collagen degradation distinguishes it from prior frameworks. The methodology is rigorous, the model is biologically grounded, and the presentation is clear and well-illustrated. Regardless of the novelty, clarity and the promising results presented by this manuscript, I have found that the authors may need to pay attention to the following issues.

Issues

1. Please clarify how the proposed finite element mechano-signaling framework improves upon or differs from earlier cartilage degeneration models, particularly those incorporating mechanical and biochemical factors.

2. Why were only day 0 and day 12 selected for experimental validation? Inclusion of intermediate time points (e.g., day 3, 6, or 9) would improve temporal resolution and model calibration.

3. Please explain the rationale for selecting ±50% parameter variation. Does this reflect biological uncertainty, empirical bounds, or a conventional modeling assumption? Were alternative distributions (e.g., uniform or log-normal) considered?

4. Figure 1: Figure 1 is somewhat blurred and visually dense. The meaning of “ABAQUS” in panel (b) is not clear. Additionally, the use of bright color scales and large arrows reduces visual clarity and focus.

5. Figure labeling and affiliations:

o Panels should be labeled with uppercase letters (A, B, C…) in the top-left corner, following journal style.

o Author affiliations should use superscript numbers (1, 2, 3…), rather than lowercase letters (a, b, c…).

Reviewer #2: Hamada et al. report in this manuscript a computational model for mechanical stress-induced cartilage damage. Their model is based on the authors’ studies on injurious cartilage tissues including the use of Fourier transform infrared microspectroscopy (FTIR) as reported earlier this year. Thus, the technical and analytical aspects as well as possible involvements of matrix metalloproteinases are not new here. The main novelty lies in computational modeling. The authors’ efforts to build a predictive model are important for future clinical applications and the development of therapeutics for osteoarthritis. It is noteworthy that the primary driver of collagen loss is attributed to the release of catabolic enzymes such as MMPs triggered by force-induced cell damage in the cartilage. Although modeling a complex system may become challenging, the authors need to address several important issues described below.

Main Issue:

I am concerned that their experimental data acquired in FTIR may not be fully utilized for modeling. In the Results section, page 16, lines 298-302, while the authors describe a result from computational simulations along with experimental chemical maps (Figs 2a and 2b), it is unclear about the reason why the area highlighted by a rectangular strip (dotted line) as an ROI (region of interest) was chosen for comparisons. The chemical/experimental maps on day 12 for either control or injury groups show highly variable and discontinuous shades of collagen distributions along the x-axis. In contrast, simulation maps show uniform distributions along the x-axis, which is probably not considered as a variable. In Methods, the authors describe that “two full-depth and 200 μm wide regions of interest (ROI) were defined in each histological section” (line 136), providing the dimension of a ROI, but not how ROIs are selected. Thus, the quantification described in Results means little. The ROI criteria should be based on objective selections of measurable parameters. The authors also need to clarify how ROIs can capture the discontinuous nature of the samples as found even in non-injurious control. Since the authors attempt to model the effect of diffusible factors generated by injurious cells in the cartilage, I feel that this is a critical issue for the authors to address properly for modeling. In the current analytical model, the chemical model is being treated as if it was a 1D model. The biomechanical and high shear strain model offers information in 2D to model the shearing force that presumably triggers cell damage and distributions of diffusible factors in a 2D plane. Therefore, there is a discrepancy between dimensions while tissue collagen contents are discontinuously localized in 2D in day-12 cartilage samples.

The issue raises the concern that the current modeling workflow appears premature in collagen distribution data interpretation. The model seems overly simplified without sufficient justification. It could be significantly strengthened by reconsidering how these special complexities are integrated.

Other issues:

1. The authors state that they did not model mechanical injuries from force, but “lesion geometry” was added to the injury model. Such lesions were based on “visual” assessments, however. If the extent (area size, shape, depth, etc.) has any impact on the injury model construction, it is highly desirable to use more objective means to identify the lesion on the images since “deformation” is indeed considered in the biomechanical continuum.

2. Another layer of the issue above is that force-induced “deformation” assumes no loss of material or change in lesion geometry over time in their model. The authors include discussion of previously observed lesion-associated rapid loss of collagen contents along authors’ simulations shown in Fig S2. Therefore, the authors need to consider the possibility that lesion geometry may not be static during the course of experiment. Evaluation of the shear strain that Day 12 samples initially received may be complicated by this difficulty to estimate the initial lesion geometry.

3. The current model uses only two time points, Day 0 and Day 12. However, additional timepoints would offer more dynamic information for better modeling for disease progression as other investigators published models on different time scales.

4. The current model reflects only isolated pieces of cartilage. The authors acknowledge the role of cytokines in promoting MMP production in the Discussion section. However, there are distinct possibilities that inflammatory cytokines may have MMP-independent effects on osteoarthritis progression in vivo. The authors should discuss the limitations of the current model.

**Have the authors made all data and (if applicable) computational code underlying the findings in their manuscript fully available?**

Reviewer #1: None

Reviewer #2: **No: ** Not yet. They will make it public after acceptance of the paper.

PLOS authors have the option to publish the peer review history of their article (what does this mean? ). If published, this will include your full peer review and any attached files.

**Do you want your identity to be public for this peer review?** For information about this choice, including consent withdrawal, please see our Privacy Policy .

Reviewer #1: No

Reviewer #2: No

**Figure resubmission:**

**Reproducibility:**



---

## [Decision Letter · Decision Letter 1]

7 Jan 2026

Dear Hamada,

We are pleased to inform you that your manuscript 'MMP release following cartilage injury leads to collagen loss in intact tissue – a computational study' has been provisionally accepted for publication in PLOS Computational Biology.

Best regards,

Arne Elofsson

Section Editor

PLOS Computational Biology

Arne Elofsson

Section Editor

PLOS Computational Biology

Reviewer's Responses to Questions

**Comments to the Authors:**

Reviewer #1: The author addressed all my concerns.

**Have the authors made all data and (if applicable) computational code underlying the findings in their manuscript fully available?**

Reviewer #1: None

PLOS authors have the option to publish the peer review history of their article (what does this mean? ). If published, this will include your full peer review and any attached files.

**Do you want your identity to be public for this peer review?** For information about this choice, including consent withdrawal, please see our Privacy Policy .

Reviewer #1: No

---

## [Editor Report · Acceptance letter]

PCOMPBIOL-D-25-01150R1

MMP release following cartilage injury leads to collagen loss in intact tissue – a computational study

Dear Dr Hamada,

I am pleased to inform you that your manuscript has been formally accepted for publication in PLOS Computational Biology. Your manuscript is now with our production department and you will be notified of the publication date in due course.

With kind regards,

Anita Estes
